# Mean teachers are better role models: Weight-averaged consistency targets improve semi-supervised deep learning results

**Antti Tarvainen**
The Curious AI Company
tarvaina@cai.fi

**Harri Valpola**
The Curious AI Company
harri@cai.fi

## Abstract

The recently proposed Temporal Ensembling has achieved state-of-the-art results in several semi-supervised learning benchmarks. It maintains an exponential moving average of label predictions on each training example, and penalizes predictions that are inconsistent with this target. However, because the targets change only once per epoch, Temporal Ensembling becomes unwieldy when learning large datasets. To overcome this problem, we propose Mean Teacher, a method that averages model weights instead of label predictions. As an additional benefit, Mean Teacher improves test accuracy and enables training with fewer labels than Temporal Ensembling. Without changing the network architecture, Mean Teacher achieves an error rate of 4.35% on SVHN with 250 labels, outperforming Temporal Ensembling trained with 1000 labels. We also show that a good network architecture is crucial to performance. Combining Mean Teacher and Residual Networks, we improve the state of the art on CIFAR-10 with 4000 labels from 10.55% to 6.28%, and on ImageNet 2012 with 10% of the labels from 35.24% to 9.11%.

## 1 Introduction

Deep learning has seen tremendous success in areas such as image and speech recognition. In order to learn useful abstractions, deep learning models require a large number of parameters, thus making them prone to over-fitting (Figure 1a). Moreover, adding high-quality labels to training data manually is often expensive. Therefore, it is desirable to use regularization methods that exploit unlabeled data effectively to reduce over-fitting in semi-supervised learning.

When a percept is changed slightly, a human typically still considers it to be the same object. Correspondingly, a classification model should favor functions that give consistent output for similar data points. One approach for achieving this is to add noise to the input of the model. To enable the model to learn more abstract invariances, the noise may be added to intermediate representations, an insight that has motivated many regularization techniques, such as Dropout [27]. Rather than minimizing the classification cost at the zero-dimensional data points of the input space, the regularized model minimizes the cost on a manifold around each data point, thus pushing decision boundaries away from the labeled data points (Figure 1b).

Since the classification cost is undefined for unlabeled examples, the noise regularization by itself does not aid in semi-supervised learning. To overcome this, the $\Gamma$ model [20] evaluates each data point with and without noise, and then applies a *consistency cost* between the two predictions. In this case, the model assumes a dual role as a *teacher* and a *student*. As a student, it learns as before; as a teacher, it generates targets, which are then used by itself as a student for learning. Since the model itself generates targets, they may very well be incorrect. If too much weight is given to the generated targets, the cost of inconsistency outweighs that of misclassification, preventing the learning of new

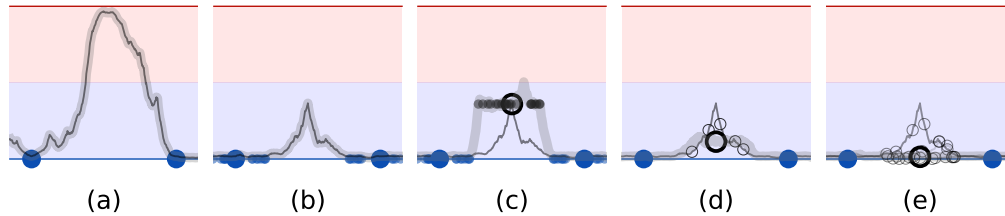

Figure 1: A sketch of a binary classification task with two labeled examples (large blue dots) and one unlabeled example, demonstrating how the choice of the unlabeled target (black circle) affects the fitted function (gray curve). **(a)** A model with no regularization is free to fit any function that predicts the labeled training examples well. **(b)** A model trained with noisy labeled data (small dots) learns to give consistent predictions around labeled data points. **(c)** Consistency to noise around unlabeled examples provides additional smoothing. For the clarity of illustration, the teacher model (gray curve) is first fitted to the labeled examples, and then left unchanged during the training of the student model. Also for clarity, we will omit the small dots in figures d and e. **(d)** Noise on the teacher model reduces the bias of the targets without additional training. The expected direction of stochastic gradient descent is towards the mean (large blue circle) of individual noisy targets (small blue circles). **(e)** An ensemble of models gives an even better expected target. Both Temporal Ensembling and the Mean Teacher method use this approach.

information. In effect, the model suffers from confirmation bias (Figure 1c), a hazard that can be mitigated by improving the quality of targets.

There are at least two ways to improve the target quality. One approach is to choose the perturbation of the representations carefully instead of barely applying additive or multiplicative noise. Another approach is to choose the teacher model carefully instead of barely replicating the student model. Concurrently to our research, Miyato et al. [15] have taken the first approach and shown that Virtual Adversarial Training can yield impressive results. We take the second approach and will show that it too provides significant benefits. To our understanding, these two approaches are compatible, and their combination may produce even better outcomes. However, the analysis of their combined effects is outside the scope of this paper.

Our goal, then, is to form a better teacher model from the student model without additional training. As the first step, consider that the softmax output of a model does not usually provide accurate predictions outside training data. This can be partly alleviated by adding noise to the model at inference time [4], and consequently a noisy teacher can yield more accurate targets (Figure 1d). This approach was used in Pseudo-Ensemble Agreement [2] and has lately been shown to work well on semi-supervised image classification [13, 22]. Laine & Aila [13] named the method the Π model; we will use this name for it and their version of it as the basis of our experiments.

The Π model can be further improved by Temporal Ensembling [13], which maintains an exponential moving average (EMA) prediction for each of the training examples. At each training step, all the EMA predictions of the examples in that minibatch are updated based on the new predictions. Consequently, the EMA prediction of each example is formed by an ensemble of the model's current version and those earlier versions that evaluated the same example. This ensembling improves the quality of the predictions, and using them as the teacher predictions improves results. However, since each target is updated only once per epoch, the learned information is incorporated into the training process at a slow pace. The larger the dataset, the longer the span of the updates, and in the case of on-line learning, it is unclear how Temporal Ensembling can be used at all. (One could evaluate all the targets periodically more than once per epoch, but keeping the evaluation span constant would require $O(n^2)$ evaluations per epoch where $n$ is the number of training examples.)

## 2 Mean Teacher

To overcome the limitations of Temporal Ensembling, we propose averaging model weights instead of predictions. Since the teacher model is an average of consecutive student models, we call this the Mean Teacher method (Figure 2). Averaging model weights over training steps tends to produce a

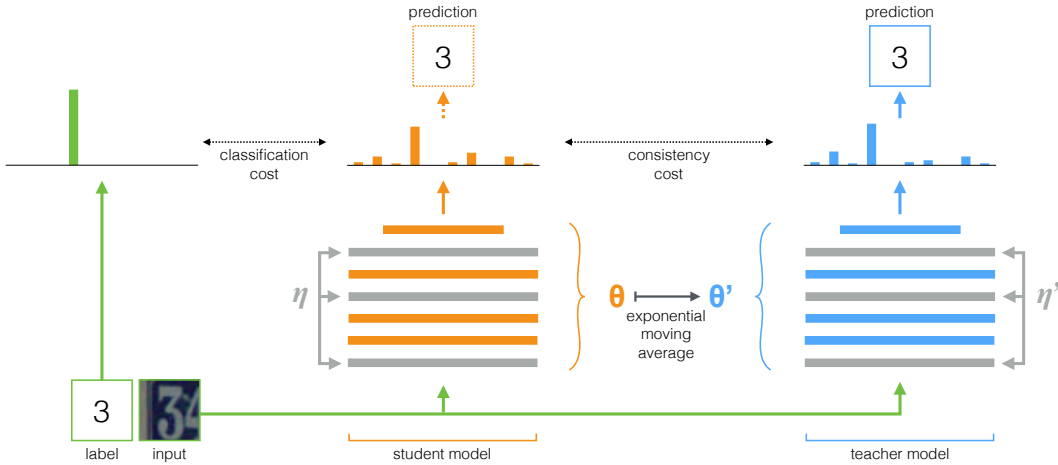

Figure 2: The Mean Teacher method. The figure depicts a training batch with a single labeled example. Both the student and the teacher model evaluate the input applying noise $(\eta, \eta')$ within their computation. The softmax output of the student model is compared with the one-hot label using classification cost and with the teacher output using consistency cost. After the weights of the student model have been updated with gradient descent, the teacher model weights are updated as an exponential moving average of the student weights. Both model outputs can be used for prediction, but at the end of the training the teacher prediction is more likely to be correct. A training step with an unlabeled example would be similar, except no classification cost would be applied.

more accurate model than using the final weights directly [18]. We can take advantage of this during training to construct better targets. Instead of sharing the weights with the student model, the teacher model uses the EMA weights of the student model. Now it can aggregate information after every step instead of every epoch. In addition, since the weight averages improve all layer outputs, not just the top output, the target model has better intermediate representations. These aspects lead to two practical advantages over Temporal Ensembling: First, the more accurate target labels lead to a faster feedback loop between the student and the teacher models, resulting in better test accuracy. Second, the approach scales to large datasets and on-line learning.

More formally, we define the consistency cost $J$ as the expected distance between the prediction of the student model (with weights $\theta$ and noise $\eta$) and the prediction of the teacher model (with weights $\theta'$ and noise $\eta'$).

$$J(\theta) = \mathbb{E}_{x,\eta',\eta} \left[ \| f(x, \theta', \eta') - f(x, \theta, \eta) \|^2 \right]$$

The difference between the $\Pi$ model, Temporal Ensembling, and Mean teacher is how the teacher predictions are generated. Whereas the $\Pi$ model uses $\theta' = \theta$, and Temporal Ensembling approximates $f(x, \theta', \eta')$ with a weighted average of successive predictions, we define $\theta'_t$ at training step $t$ as the EMA of successive $\theta$ weights:

$$\theta'_t = \alpha \theta'_{t-1} + (1 - \alpha)\theta_t$$

where $\alpha$ is a smoothing coefficient hyperparameter. An additional difference between the three algorithms is that the $\Pi$ model applies training to $\theta'$ whereas Temporal Ensembling and Mean Teacher treat it as a constant with regards to optimization.

We can approximate the consistency cost function $J$ by sampling noise $\eta, \eta'$ at each training step with stochastic gradient descent. Following Laine & Aila [13], we use mean squared error (MSE) as the consistency cost in most of our experiments.

Table 1: Error rate percentage on SVHN over 10 runs (4 runs when using all labels). We use exponential moving average weights in the evaluation of all our models. All the methods use a similar 13-layer ConvNet architecture. See Table 5 in the Appendix for results without input augmentation.

|  | 250 labels 73257 images | 500 labels 73257 images | 1000 labels 73257 images | 73257 labels 73257 images |
|---|---|---|---|---|
| GAN [24] |  | $18.44 \pm 4.8$ | $8.11 \pm 1.3$ |  |
| Π model [13] |  | $6.65 \pm 0.53$ | $4.82 \pm 0.17$ | $2.54 \pm 0.04$ |
| Temporal Ensembling [13] |  | $5.12 \pm 0.13$ | $4.42 \pm 0.16$ | $2.74 \pm 0.06$ |
| VAT+EntMin [15] |  |  | **3.86** |  |
| Supervised-only | $27.77 \pm 3.18$ | $16.88 \pm 1.30$ | $12.32 \pm 0.95$ | $2.75 \pm 0.10$ |
| Π model | $9.69 \pm 0.92$ | $6.83 \pm 0.66$ | $4.95 \pm 0.26$ | $2.50 \pm 0.07$ |
| Mean Teacher | **$4.35 \pm 0.50$** | **$4.18 \pm 0.27$** | $3.95 \pm 0.19$ | **$2.50 \pm 0.05$** |

Table 2: Error rate percentage on CIFAR-10 over 10 runs (4 runs when using all labels).

|  | 1000 labels 50000 images | 2000 labels 50000 images | 4000 labels 50000 images | 50000 labels 50000 images |
|---|---|---|---|---|
| GAN [24] |  |  | $18.63 \pm 2.32$ |  |
| Π model [13] |  |  | $12.36 \pm 0.31$ | $5.56 \pm 0.10$ |
| Temporal Ensembling [13] |  |  | $12.16 \pm 0.31$ | **$5.60 \pm 0.10$** |
| VAT+EntMin [15] |  |  | **10.55** |  |
| Supervised-only | $46.43 \pm 1.21$ | $33.94 \pm 0.73$ | $20.66 \pm 0.57$ | $5.82 \pm 0.15$ |
| Π model | $27.36 \pm 1.20$ | $18.02 \pm 0.60$ | $13.20 \pm 0.27$ | $6.06 \pm 0.11$ |
| Mean Teacher | **$21.55 \pm 1.48$** | **$15.73 \pm 0.31$** | $12.31 \pm 0.28$ | $5.94 \pm 0.15$ |

## 3 Experiments

To test our hypotheses, we first replicated the Π model [13] in TensorFlow [1] as our baseline. We then modified the baseline model to use weight-averaged consistency targets. The model architecture is a 13-layer convolutional neural network (ConvNet) with three types of noise: random translations and horizontal flips of the input images, Gaussian noise on the input layer, and dropout applied within the network. We use mean squared error as the consistency cost and ramp up its weight from 0 to its final value during the first 80 epochs. The details of the model and the training procedure are described in Appendix B.1.

### 3.1 Comparison to other methods on SVHN and CIFAR-10

We ran experiments using the Street View House Numbers (SVHN) and CIFAR-10 benchmarks [16]. Both datasets contain 32x32 pixel RGB images belonging to ten different classes. In SVHN, each example is a close-up of a house number, and the class represents the identity of the digit at the center of the image. In CIFAR-10, each example is a natural image belonging to a class such as horses, cats, cars and airplanes. SVHN contains of 73257 training samples and 26032 test samples. CIFAR-10 consists of 50000 training samples and 10000 test samples.

Tables 1 and 2 compare the results against recent state-of-the-art methods. All the methods in the comparison use a similar 13-layer ConvNet architecture. Mean Teacher improves test accuracy over the Π model and Temporal Ensembling on semi-supervised SVHN tasks. Mean Teacher also improves results on CIFAR-10 over our baseline Π model.

The recently published version of Virtual Adversarial Training by Miyato et al. [15] performs even better than Mean Teacher on the 1000-label SVHN and the 4000-label CIFAR-10. As discussed in the introduction, VAT and Mean Teacher are complimentary approaches. Their combination may yield better accuracy than either of them alone, but that investigation is beyond the scope of this paper.

Table 3: Error percentage over 10 runs on SVHN with extra unlabeled training data.

|  | 500 labels 73257 images | 500 labels 173257 images | 500 labels 573257 images |
|---|---|---|---|
| Π model (ours) | $6.83 \pm 0.66$ | $4.49 \pm 0.27$ | $3.26 \pm 0.14$ |
| Mean Teacher | $\mathbf{4.18 \pm 0.27}$ | $\mathbf{3.02 \pm 0.16}$ | $\mathbf{2.46 \pm 0.06}$ |

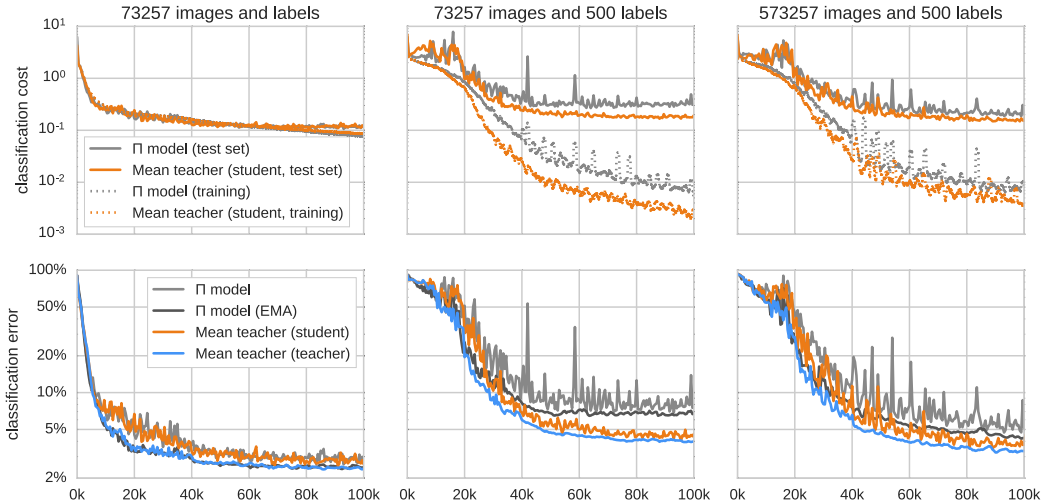

Figure 3: Smoothened classification cost (top) and classification error (bottom) of Mean Teacher and our baseline Π model on SVHN over the first 100000 training steps. In the upper row, the training classification costs are measured using only labeled data.

## 3.2 SVHN with extra unlabeled data

Above, we suggested that Mean Teacher scales well to large datasets and on-line learning. In addition, the SVHN and CIFAR-10 results indicate that it uses unlabeled examples efficiently. Therefore, we wanted to test whether we have reached the limits of our approach.

Besides the primary training data, SVHN includes also an extra dataset of 531131 examples. We picked 500 samples from the primary training as our labeled training examples. We used the rest of the primary training set together with the extra training set as unlabeled examples. We ran experiments with Mean Teacher and our baseline Π model, and used either 0, 100000 or 500000 extra examples. Table 3 shows the results.

## 3.3 Analysis of the training curves

The training curves on Figure 3 help us understand the effects of using Mean Teacher. As expected, the EMA-weighted models (blue and dark gray curves in the bottom row) give more accurate predictions than the bare student models (orange and light gray) after an initial period.

Using the EMA-weighted model as the teacher improves results in the semi-supervised settings. There appears to be a virtuous feedback cycle of the teacher (blue curve) improving the student (orange) via the consistency cost, and the student improving the teacher via exponential moving averaging. If this feedback cycle is detached, the learning is slower, and the model starts to overfit earlier (dark gray and light gray).

Mean Teacher helps when labels are scarce. When using 500 labels (middle column) Mean Teacher learns faster, and continues training after the Π model stops improving. On the other hand, in the all-labeled case (left column), Mean Teacher and the Π model behave virtually identically.

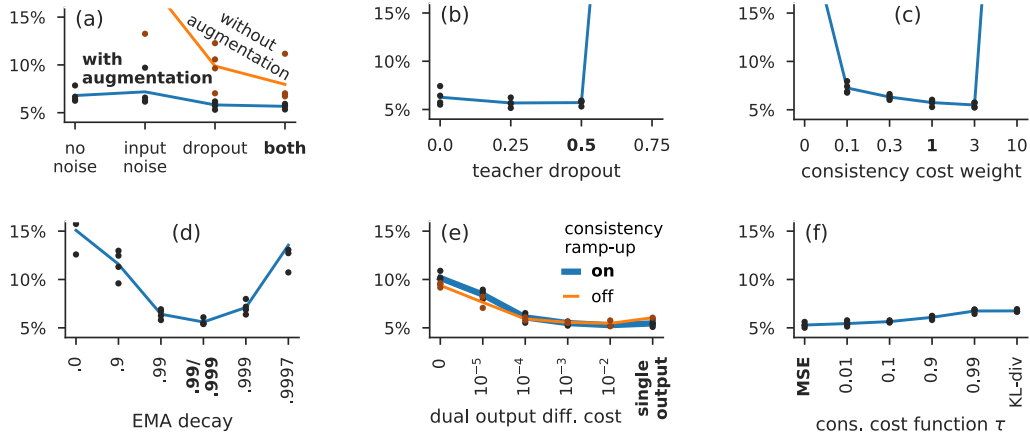

Figure 4: Validation error on 250-label SVHN over four runs per hyperparameter setting and their means. In each experiment, we varied one hyperparameter, and used the evaluation run hyperparameters of Table 1 for the rest. The hyperparameter settings used in the evaluation runs are marked with the bolded font weight. See the text for details.

Mean Teacher uses unlabeled training data more efficiently than the $\Pi$ model, as seen in the middle column. On the other hand, with 500k extra unlabeled examples (right column), $\Pi$ model keeps improving for longer. Mean Teacher learns faster, and eventually converges to a better result, but the sheer amount of data appears to offset $\Pi$ model's worse predictions.

### 3.4 Ablation experiments

To assess the importance of various aspects of the model, we ran experiments on SVHN with 250 labels, varying one or a few hyperparameters at a time while keeping the others fixed.

**Removal of noise** (Figures 4(a) and 4(b)). In the introduction and Figure 1, we presented the hypothesis that the $\Pi$ model produces better predictions by adding noise to the model on both sides. But after the addition of Mean Teacher, is noise still needed? Yes. We can see that either input augmentation or dropout is necessary for passable performance. On the other hand, input noise does not help when augmentation is in use. Dropout on the teacher side provides only a marginal benefit over just having it on the student side, at least when input augmentation is in use.

**Sensitivity to EMA decay and consistency weight** (Figures 4(c) and 4(d)). The essential hyperparameters of the Mean Teacher algorithm are the consistency cost weight and the EMA decay $\alpha$. How sensitive is the algorithm to their values? We can see that in each case the good values span roughly an order of magnitude and outside these ranges the performance degrades quickly. Note that EMA decay $\alpha = 0$ makes the model a variation of the $\Pi$ model, although somewhat inefficient one because the gradients are propagated through only the student path. Note also that in the evaluation runs we used EMA decay $\alpha = 0.99$ during the ramp-up phase, and $\alpha = 0.999$ for the rest of the training. We chose this strategy because the student improves quickly early in the training, and thus the teacher should forget the old, inaccurate, student weights quickly. Later the student improvement slows, and the teacher benefits from a longer memory.

**Decoupling classification and consistency** (Figure 4(e)). The consistency to teacher predictions may not necessarily be a good proxy for the classification task, especially early in the training. So far our model has strongly coupled these two tasks by using the same output for both. How would decoupling the tasks change the performance of the algorithm? To investigate, we changed the model to have two top layers and produce two outputs. We then trained one of the outputs for classification and the other for consistency. We also added a mean squared error cost between the output logits, and then varied the weight of this cost, allowing us to control the strength of the coupling. Looking at the results (reported using the EMA version of the classification output), we can see that the strongly coupled version performs well and the too loosely coupled versions do not. On the other hand, a moderate decoupling seems to have the benefit of making the consistency ramp-up redundant.

Table 4: Error rate percentage of ResNet Mean Teacher compared to the state of the art. We report the test results from 10 runs on CIFAR-10 and validation results from 2 runs on ImageNet.

|  | CIFAR-10 4000 labels | ImageNet 2012 10% of the labels |
|---|---|---|
| State of the art | 10.55 [15] | $35.24 \pm 0.90$ [19] |
| ConvNet Mean Teacher | $12.31 \pm 0.28$ | |
| ResNet Mean Teacher | **$6.28 \pm 0.15$** | **$9.11 \pm 0.12$** |
| State of the art using all labels | 2.86 [5] | 3.79 [10] |

**Changing from MSE to KL-divergence** (Figure 4(f)) Following Laine & Aila [13], we use mean squared error (MSE) as our consistency cost function, but KL-divergence would seem a more natural choice. Which one works better? We ran experiments with instances of a cost function family ranging from MSE ($\tau = 0$ in the figure) to KL-divergence ($\tau = 1$), and found out that in this setting MSE performs better than the other cost functions. See Appendix C for the details of the cost function family and for our intuition about why MSE performs so well.

## 3.5  Mean Teacher with residual networks on CIFAR-10 and ImageNet

In the experiments above, we used a traditional 13-layer convolutional architecture (ConvNet), which has the benefit of making comparisons to earlier work easy. In order to explore the effect of the model architecture, we ran experiments using a 12-block (26-layer) Residual Network [8] (ResNet) with Shake-Shake regularization [5] on CIFAR-10. The details of the model and the training procedure are described in Appendix B.2. As shown in Table 4, the results improve remarkably with the better network architecture.

To test whether the methods scales to more natural images, we ran experiments on Imagenet 2012 dataset [21] using 10% of the labels. We used a 50-block (152-layer) ResNeXt architecture [32], and saw a clear improvement over the state of the art. As the test set is not publicly available, we measured the results using the validation set.

## 4  Related work

Noise regularization of neural networks was proposed by Sietsma & Dow [25]. More recently, several types of perturbations have been shown to regularize intermediate representations effectively in deep learning. Adversarial Training [6] changes the input slightly to give predictions that are as different as possible from the original predictions. Dropout [27] zeroes random dimensions of layer outputs. Dropconnect [30] generalizes Dropout by zeroing individual weights instead of activations. Stochastic Depth [11] drops entire layers of residual networks, and Swapout [26] generalizes Dropout and Stochastic Depth. Shake-shake regularization [5] duplicates residual paths and samples a linear combination of their outputs independently during forward and backward passes.

Several semi-supervised methods are based on training the model predictions to be consistent to perturbation. The Denoising Source Separation framework (DSS) [28] uses denoising of latent variables to learn their likelihood estimate. The $\Gamma$ variant of Ladder Network [20] implements DSS with a deep learning model for classification tasks. It produces a noisy student predictions and clean teacher predictions, and applies a denoising layer to predict teacher predictions from the student predictions. The $\Pi$ model [13] improves the $\Gamma$ model by removing the explicit denoising layer and applying noise also to the teacher predictions. Similar methods had been proposed already earlier for linear models [29] and deep learning [2]. Virtual Adversarial Training [15] is similar to the $\Pi$ model but uses adversarial perturbation instead of independent noise.

The idea of a teacher model training a student is related to model compression [3] and distillation [9]. The knowledge of a complicated model can be transferred to a simpler model by training the simpler model with the softmax outputs of the complicated model. The softmax outputs contain more information about the task than the one-hot outputs, and the requirement of representing this

knowledge regularizes the simpler model. Besides its use in model compression, distillation can be used to harden trained models against adversarial attacks [17]. The difference between distillation and consistency regularization is that distillation is performed after training whereas consistency regularization is performed on training time.

Consistency regularization can be seen as a form of label propagation [33]. Training samples that resemble each other are more likely to belong to the same class. Label propagation takes advantage of this assumption by pushing label information from each example to examples that are near it according to some metric. Label propagation can also be applied to deep learning models [31]. However, ordinary label propagation requires a predefined distance metric in the input space. In contrast, consistency targets employ a learned distance metric implied by the abstract representations of the model. As the model learns new features, the distance metric changes to accommodate these features. Therefore, consistency targets guide learning in two ways. On the one hand they spread the labels according to the current distance metric, and on the other hand, they aid the network learn a better distance metric.

## 5    Conclusion

Temporal Ensembling, Virtual Adversarial Training and other forms of consistency regularization have recently shown their strength in semi-supervised learning. In this paper, we propose Mean Teacher, a method that averages model weights to form a target-generating teacher model. Unlike Temporal Ensembling, Mean Teacher works with large datasets and on-line learning. Our experiments suggest that it improves the speed of learning and the classification accuracy of the trained network. In addition, it scales well to state-of-the-art architectures and large image sizes.

The success of consistency regularization depends on the quality of teacher-generated targets. If the targets can be improved, they should be. Mean Teacher and Virtual Adversarial Training represent two ways of exploiting this principle. Their combination may yield even better targets. There are probably additional methods to be uncovered that improve targets and trained models even further.

## Acknowledgements

We thank Samuli Laine and Timo Aila for fruitful discussions about their work, and Phil Bachman and Colin Raffel for corrections to the pre-print version of this paper. We also thank everyone at The Curious AI Company for their help, encouragement, and ideas.

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
