[Supplementary Material · mean-teacher-supplementary.pdf]

# Appendix

## A  Results without input augmentation

See table 5 for the results without input augmentation.

Table 5: Error rate percentage on SVHN and CIFAR-10 over 10 runs, including the results without input augmentation. We use exponential moving average weights in the evaluation of all our models. All the comparison methods use a 13-layer ConvNet architecture similar to ours and augmentation similar to ours, expect GAN, which does not use augmentation.

| SVHN | 250 labels | 500 labels | 1000 labels | all labels[a] |
|---|---|---|---|---|
| GAN[b] | | $18.44 \pm 4.8$ | $8.11 \pm 1.3$ | |
| Π model[c] | | $6.65 \pm 0.53$ | $4.82 \pm 0.17$ | $2.54 \pm 0.04$ |
| Temporal Ensembling[c] | | $5.12 \pm 0.13$ | $4.42 \pm 0.16$ | $2.74 \pm 0.06$ |
| VAT+EntMin[d] | | | **3.86** | |
| Ours | | | | |
|     Supervised-only[e] | $27.77 \pm 3.18$ | $16.88 \pm 1.30$ | $12.32 \pm 0.95$ | $2.75 \pm 0.10$ |
|     Π model | $9.69 \pm 0.92$ | $6.83 \pm 0.66$ | $4.95 \pm 0.26$ | $2.50 \pm 0.07$ |
|     Mean Teacher | $\mathbf{4.35 \pm 0.50}$ | $\mathbf{4.18 \pm 0.27}$ | $3.95 \pm 0.19$ | $\mathbf{2.50 \pm 0.05}$ |
| Without augmentation | | | | |
|     Supervised-only[e] | $36.26 \pm 3.83$ | $19.68 \pm 1.03$ | $14.15 \pm 0.87$ | $3.04 \pm 0.04$ |
|     Π model | $10.36 \pm 0.94$ | $7.01 \pm 0.29$ | $5.73 \pm 0.16$ | $2.75 \pm 0.08$ |
|     Mean Teacher | $5.85 \pm 0.62$ | $5.45 \pm 0.14$ | $5.21 \pm 0.21$ | $2.77 \pm 0.09$ |

| CIFAR-10 | 1000 labels | 2000 labels | 4000 labels | all labels[a] |
|---|---|---|---|---|
| GAN[b] | | | $18.63 \pm 2.32$ | |
| Π model[c] | | | $12.36 \pm 0.31$ | $\mathbf{5.56 \pm 0.10}$ |
| Temporal Ensembling[c] | | | $12.16 \pm 0.31$ | $5.60 \pm 0.10$ |
| VAT+EntMin[d] | | | $10.55$ | |
| Ours | | | | |
|     Supervised-only[e] | $46.43 \pm 1.21$ | $33.94 \pm 0.73$ | $20.66 \pm 0.57$ | $5.82 \pm 0.15$ |
|     Π model | $27.36 \pm 1.20$ | $18.02 \pm 0.60$ | $13.20 \pm 0.27$ | $6.06 \pm 0.11$ |
|     Mean Teacher | $21.55 \pm 1.48$ | $\mathbf{15.73 \pm 0.31}$ | $12.31 \pm 0.28$ | $5.94 \pm 0.15$ |
|     Mean Teacher ResNet | $\mathbf{10.08 \pm 0.41}$ | | $\mathbf{6.28 \pm 0.15}$ | |
| Without augmentation | | | | |
|     Supervised-only[e] | $48.38 \pm 1.07$ | $36.07 \pm 0.90$ | $24.47 \pm 0.50$ | $7.43 \pm 0.06$ |
|     Π model | $32.18 \pm 1.33$ | $23.92 \pm 1.07$ | $17.08 \pm 0.32$ | $7.00 \pm 0.20$ |
|     Mean Teacher | $30.62 \pm 1.13$ | $23.14 \pm 0.46$ | $17.74 \pm 0.30$ | $7.21 \pm 0.24$ |

[a] 4 runs    [b] Salimans et al. [25]    [c] Laine & Aila [13]    [d] Miyato et al. [16]
[e] Only labeled examples and only classification cost

## B  Experimental setup

Source code for the experiments is available at `https://github.com/CuriousAI/mean-teacher`.

### B.1  Convolutional network models

We replicated the Π model of Laine & Aila [13] in TensorFlow [1], and added support for Mean Teacher training. We modified the model slightly to match the requirements of the experiments, as described in subsections B.1.1 and B.1.2. The difference between the original Π model described by Laine & Aila [13] and our baseline Π model thus depends on the experiment. The difference between

Table 6: The convolutional network architecture we used in the experiments.

| Layer | Hyperparameters |
|---|---|
| Input | $32 \times 32$ RGB image |
| Translation | Randomly $\{\Delta x, \Delta y\} \sim [-2, 2]$ |
| Horizontal flip[a] | Randomly $p = 0.5$ |
| Gaussian noise | $\sigma = 0.15$ |
| Convolutional | 128 filters, $3 \times 3$, *same* padding |
| Convolutional | 128 filters, $3 \times 3$, *same* padding |
| Convolutional | 128 filters, $3 \times 3$, *same* padding |
| Pooling | Maxpool $2 \times 2$ |
| Dropout | $p = 0.5$ |
| Convolutional | 256 filters, $3 \times 3$, *same* padding |
| Convolutional | 256 filters, $3 \times 3$, *same* padding |
| Convolutional | 256 filters, $3 \times 3$, *same* padding |
| Pooling | Maxpool $2 \times 2$ |
| Dropout | $p = 0.5$ |
| Convolutional | 512 filters, $3 \times 3$, *valid* padding |
| Convolutional | 256 filters, $1 \times 1$, *same* padding |
| Convolutional | 128 filters, $1 \times 1$, *same* padding |
| Pooling | Average pool ($6 \times 6 \rightarrow 1 \times 1$ pixels) |
| Softmax | Fully connected $128 \rightarrow 10$ |

[a] Not applied on SVHN experiments

our baseline $\Pi$ model and our Mean Teacher model is whether the teacher weights are identical to the student weights or an EMA of the student weights. In addition, the $\Pi$ models (both the original and ours) backpropagate gradients to both sides of the model whereas Mean Teacher applies them only to the student side.

Table 6 describes the architecture of the convolutional network. We applied mean-only batch normalization and weight normalization [24] on convolutional and softmax layers. We used Leaky ReLu [15] with $\alpha = 0.1$ as the nonlinearity on each of the convolutional layers.

We used cross-entropy between the student softmax output and the one-hot label as the classification cost, and the mean square error between the student and teacher softmax outputs as the consistency cost. The total cost was the weighted sum of these costs, where the weight of classification cost was the expected number of labeled examples per minibatch, subject to the ramp-ups described below.

We trained the network with minibatches of size 100. We used Adam Optimizer [12] for training with learning rate 0.003 and parameters $\beta_1 = 0.9$, $\beta_2 = 0.999$, and $\varepsilon = 10^{-8}$. In our baseline $\Pi$ model we applied gradients through both teacher and student sides of the network. In Mean teacher model, the teacher model parameters were updated after each training step using an EMA with $\alpha = 0.999$. These hyperparameters were subject to the ramp-ups and ramp-downs described below.

We applied a ramp-up period of 40000 training steps at the beginning of training. The consistency cost coefficient and the learning rate were ramped up from 0 to their maximum values, using a sigmoid-shaped function $e^{-5(1-x)^2}$, where $x \in [0, 1]$.

We used different training settings in different experiments. In the CIFAR-10 experiment, we matched the settings of Laine & Aila [13] as closely as possible. In the SVHN experiments, we diverged from Laine & Aila [13] to accommodate for the sparsity of labeled data. Table 7 summarizes the differences between our experiments.

### B.1.1 ConvNet on CIFAR-10

We normalized the input images with ZCA based on training set statistics.

For sampling minibatches, the labeled and unlabeled examples were treated equally, and thus the number of labeled examples varied from minibatch to minibatch.

We applied a ramp-down for the last 25000 training steps. The learning rate coefficient was ramped down to 0 from its maximum value. Adam $\beta_1$ was ramped down to $0.5$ from its maximum value. The ramp-downs were performed using sigmoid-shaped function $1 - e^{-12.5x^2}$, where $x \in [0, 1]$. These ramp-downs did not improve the results, but were used to stay as close as possible to the settings of Laine & Aila [13].

### B.1.2   ConvNet on SVHN

We normalized the input images to have zero mean and unit variance.

When doing semi-supervised training, we used 1 labeled example and 99 unlabeled examples in each mini-batch. This was important to speed up training when using extra unlabeled data. After all labeled examples had been used, they were shuffled and reused. Similarly, after all unlabeled examples had been used, they were shuffled and reused.

We applied different values for Adam $\beta_2$ and EMA decay rate during the ramp-up period and the rest of the training. Both of the values were $0.99$ during the first 40000 steps, and $0.999$ afterwards. This helped the 250-label case converge reliably.

We trained the network for 180000 steps when not using extra unlabeled examples, for 400000 steps when using 100k extra unlabeled examples, and for 600000 steps when using 500k extra unlabeled examples.

### B.1.3   The baseline ConvNet models

For training the supervised-only and Π model baselines we used the same hyperparameters as for training the Mean Teacher, except we stopped training earlier to prevent over-fitting. For supervised-only runs we did not include any unlabeled examples and did not apply the consistency cost.

We trained the supervised-only model on CIFAR-10 for 7500 steps when using 1000 images, for 15000 steps when using 2000 images, for 30000 steps when using 4000 images and for 150000 steps when using all images. We trained it on SVHN for 40000 steps when using 250, 500 or 1000 labels, and for 180000 steps when using all labels.

We trained the Π model on CIFAR-10 for 60000 steps when using 1000 labels, for 100000 steps when using 2000 labels, and for 180000 steps when using 4000 labels or all labels. We trained it on SVHN for 100000 steps when using 250 labels, and for 180000 steps when using 500, 1000, or all labels.

## B.2   Residual network models

We implemented our residual network experiments in PyTorch[1]. We used different architectures for our CIFAR-10 and ImageNet experiments.

### B.2.1   ResNet on CIFAR-10

For CIFAR-10, we replicated the 26-2x96d Shake-Shake regularized architecture described in [5], and consisting of 4+4+4 residual blocks.

We trained the network on 4 GPUs using minibatches of 512 images, 124 of which were labeled. We sampled the images in the same way as described in the SVHN experiments above. We augmented the input images with 4x4 random translations (reflecting the pixels at borders when necessary) and random horizontal flips. (Note that following [5] we used a larger translation size than on our earlier experiments.) We normalized the images to have channel-wise zero mean and unit variance over training data.

We trained the network using stochastic gradient descent with initial learning rate 0.2 and Nesterov momentum 0.9. We trained for 180 epochs, decaying the learning rate with cosine annealing [14]

Table 7: Differences in training settings between the ConvNet experiments

| Aspect | semi-supervised SVHN | supervised SVHN | semi-supervised CIFAR-10 |
|---|---|---|---|
| image pre-processing | zero mean, unit variance | zero mean, unit variance | ZCA |
| image augmentation | translation | translation | translation + horizontal flip |
| number of labeled examples per minibatch | 1 | 100 | varying |
| training steps | 180000-600000 | 180000 | 150000 |
| Adam $\beta_2$ during and after ramp-up | 0.99, 0.999 | 0.99, 0.999 | 0.999, 0.999 |
| EMA decay rate during and after ramp-up | 0.99, 0.999 | 0.99, 0.999 | 0.999, 0.999 |
| Ramp-downs | No | No | Yes |

so that it would have reached zero after 210 epochs. We define epoch as one pass through all the unlabeled examples – each labeled example was included many times in one such epoch.

We used a total cost function consisting of classification cost and three other costs: We used the dual output trick described in subsection 3.4 and Figure 4(e) with MSE cost between logits with coefficient 0.01. This simplified other hyperparameter choices and improved the results. We used MSE consistency cost with coefficient ramping up from 0 to 100.0 during the first 5 epochs, using the same sigmoid ramp-up shape as in the experiments above. We also used an L2 weight decay with coefficient 2e-4. We used EMA decay value 0.97.

### B.2.2 ResNet on ImageNet

On our ImageNet evaluation runs, we used a 152-layer ResNeXt architecture [33] consisting of 3+8+36+3 residual blocks, with 32 groups of 4 channels on the first block.

We trained the network on 10 GPUs using minibatches of 400 images, 200 of which were labeled. We sampled the images in the same way as described in the SVHN experiments above. Following [10], we randomly augmented images using a 10 degree rotation, a crop with aspect ratio between 3/4 and 4/3 resized to 224x224 pixels, a random horizontal flip and a color jitter. We then normalized images to have channel-wise zero mean and unit variance over training data.

We trained the network using stochastic gradient descent with maximum learning rate 0.25 and Nesterov momentum 0.9. We ramped up the learning rate linearly during the first two epochs from 0.1 to 0.25. We trained for 60 epochs, decaying the learning rate with cosine annealing so that it would have reached zero after 75 epochs.

We used a total cost function consisting of classification cost and three other costs: We used the dual output trick described in subsection 3.4 and Figure 4(e) with MSE cost between logits with coefficient 0.01. We used a KL-divergence consistency cost with coefficient ramping up from 0 to 10.0 during the first 5 epochs, using the same sigmoid ramp-up shape as in the experiments above. We also used an L2 weight decay with coefficient 5e-5. We used EMA decay value 0.9997.

Figure 5: Copy of Figure 4(f) in the main text. Validation error on 250-label SVHN over four runs and their mean, when varying the consistency cost shape hyperparameter $\tau$ between mean squared error ($\tau = 0$) and KL-divergence ($\tau = 1$).

## B.3  Use of training, validation and test data

In the development phase of our work with CIFAR-10 and SVHN datasets, we separated 10% of training data into a validation set. We removed randomly most of the labels from the remaining training data, retaining an equal number of labels from each class. We used a different set of labels for each of the evaluation runs. We retained labels in the validation set to enable exploration of the results. In the final evaluation phase we used the entire training set, including the validation set but with labels removed.

On a real-world use case we would not possess a large fully-labeled validation set. However, this setup is useful in a research setting, since it enables a more thorough analysis of the results. To the best of our knowledge, this is the common practice when carrying out research on semi-supervised learning. By retaining the hyperparameters from previous work where possible we decreased the chance of over-fitting our results to validation labels.

In the ImageNet experiments we removed randomly most of the labels from the training set, retaining an equal number of labels from each class. For validation we used the given validation set without modifications. We used a different set of training labels for each of the evaluation runs and evaluated the results against the validation set.

## C  Varying between mean squared error and KL-divergence

As mentioned in subsection 3.4, we ran an experiment varying the consistency cost function between MSE and KL-divergence (reproduced in Figure 5). The exact consistency function we used was

$$C_\tau(p,q) = Z_\tau D_{\mathrm{KL}}(p_\tau \| q_\tau), \quad \text{where} \quad Z_\tau = \frac{2}{N^2 \tau^2}, \quad p_\tau = \tau p + \frac{1-\tau}{N}, \quad q_\tau = \tau q + \frac{1-\tau}{N},$$

$\tau \in (0,1]$ and $N$ is the number of classes. Taking the Taylor expansion we get

$$D_{\mathrm{KL}}(p_i \| q_i) = \sum_i \frac{1}{2} \tau^2 N (p_i - q_i)^2 + O\left(N^2 \tau^3\right)$$

where the zeroth- and first-order terms vanish. Consequently,

$$C_\tau(p,q) \to \frac{1}{N} \sum_i (p_i - q_i)^2 \qquad \text{when } \tau \to 0$$

$$C_\tau(p,q) = \frac{2}{N^2} D_{\mathrm{KL}}\left(p \| q\right) \qquad \text{when } \tau = 1.$$

The results in Figure 5 show that MSE performs better than KL-divergence or $C_\tau$ with any $\tau$. We also tried other consistency cost weights with KL-divergence and did not reach the accuracy of MSE.

The exact reason why MSE performs better than KL-divergence remains unclear, but the form of $C_\tau$ may help explain it. Modern neural network architectures tend to produce accurate but overly confident predictions [7]. We can assume that the true labels are accurate, but we should discount the confidence of the teacher predictions. We can do that by having $\tau = 1$ for the classification cost and $\tau < 1$ for the consistency cost. Then $p_\tau$ and $q_\tau$ discount the confidence of the approximations while $Z_\tau$ keeps gradients large enough to provide a useful training signal. However, we did not perform experiments to validate this explanation.

## Footnotes

[1]https://github.com/pytorch/pytorch