[Reviews · NeurIPS 2017]

Reviewer 1



This work describes a simple approach to improve semi-supervised learning by training a student to be consistent with the predictions of a teacher that is simply a moving average of the student. The approach and motivation is easy to understand and the paper clearly written. The results are quite impressive. Figure 1 is quite unclear. Since in 1(a) the model (gray curve) has fit to the unlabeled target, then the classifier must be an unsupervised classifier but the caption doesn't indicate that. The \Pi model is also repeatedly mentioned in the text and the results table but there is no clear description of it. In line 74, is the mean square error applied on the logits of the model? The text should also describe the architecture of the CNN used to produce the supervised results. Am I right to assume table 1 doesn't use any unlabelled data? What was the value of \alpha that worked in practice? Minor comment: Line 106 is badly phrased.

Reviewer 2



The paper proposes a new method for using unlabeled data in semi-supervised learning. The idea is to construct a teacher network from student network during training by using an exponentially decaying moving average of the weights of the student network, updating after each batch. This is inspired by previous work that uses a temporal ensemble of the softmax outputs, and aims to reduce the variance of the targets during training. Noise of various forms is added to both labelled and unlabeled examples, and a L2 penalty is added to encourage the student outputs to be consistent with the teachers. As the authors mention, this acts as a kind of soft adaptive label propagation mechanism. The advantage of their approach over temporal ensembling is that it can be used in the online setting. It also appears to converge faster and gives better results in the settings with very few labels. Strengths - Can be used in the online learning scenario. - Results are comparable with other state of the art on CIFAR-10 and SVNH. - The approach could easily be combined with other techniques for further improvement. - Experiment configuration details given in full in the appendix. - Simple approach and should be widely applicable. Weaknesses - More evaluation would have been welcome, especially on CIFAR-10 in the full label and lower label scenarios. - The CIFAR-10 results are a little disappointing with respect to temporal ensembles (although the results are comparable and the proposed approach has other advantages) - An evaluation on the more challenging STL-10 dataset would have been welcome. Comments - The SVNH evaluation suggests that the model is better than pi an temporal ensembling especially in the low-label scenario. With this in mind, it would have been nice to see if you can confirm this on CIFAR-10 too (i.e. show results on CIFAR-10 with less labels) - I would would have like to have seen what the CIFAR-10 performance looks like with all labels included. - It would be good to include in the left graph in fig 3 the learning curve for a model without any mean teacher or pi regularization for comparison, to see if mean teacher accelerates learning or slows it down. - I'd be interested to see if the exponential moving average of the weights provides any benefit on it's own, without the additional consistency cost.

Reviewer 3



Summary- This paper shows that an exponential moving average (in parameter space) of models produced during SGD training of neural networks can be used as a good teacher model for the student model which corresponds to the last (current) SGD iterate. The teacher model is used to provide a target (softmax probabilities) which is regressed to by the student model on both labelled and unlabelled data. This additional loss serves as a regularizer. In other words, the model is being trained to be consistent with its Polyak-averaged version. Furthermore, noise is injected into both the student and teacher models to increase the regularization effect (similar motivation as dropout and other related methods). This is shown to be more effective than (1) Just using noise (and no moving average), and (2) using a moving average over softmax probabilities (not parameters) which is updated once every epoch. Strengths- - The proposed teacher model is a convenient one to maintain. It just requires keeping a moving average. - The additional computational cost is just that of a forward prop and benefits seem to be worth it (at least for small amounts of labelled data). - The supplementary material provides enough experimental details. Weakness- - Comparison to other semi-supervised approaches : Other approaches such as variants of Ladder networks would be relevant models to compare to. Questions/Comments- - In Table 3, what is the difference between \Pi and \Pi (ours) ? - In Table 3, is EMA-weighting used for other baseline models ("Supervised", \Pi, etc) ? To ensure a fair comparison, it would be good to know that all the models being compared to make use of the EMA benefits. - The proposed model benefits from two factors : noise and keeping an exponential moving average. It would be good to see how much each factor contributes on its own. The \Pi model captures just the noise part, so it would be useful to know how much gain can be obtained by just using a noise-free exponential moving average. - If averaging in parameter space is being used, it seems that it should be possible to apply the consistency cost in the intermediate layers of the model as well. That could potentially provide a richer consistency gradient. Was this tried ? Minor comments and typos- - In the abstract : "The recently proposed Temporal Ensembling has ... ": Please cite. - "when learning large datasets." -> "when learning on large datasets." - "zero-dimensional data points of the input space": It may not be accurate to say that the data points are zero-dimensional. - "barely applying", " barely replicating" : "barely" -> "merely" - "softmax output of a model does not provide a good Bayesian approximation outside training data". Bayesian approximation to what ? Please explain. Any model will have some more generalization error outside training data. Is there another source of error being referred to here ? Overall- The paper proposes a simple and effective way of using unlabelled data and improving generalization with labelled data. The most attractive property is probably the low overhead of using this in practice, so it is quite likely that this approach could be impactful and widely used.